Evolutionary drivers of the hump-shaped latitudinal gradient of benthic polychaete species richness along the Southeastern Pacific coast

Moreno Rodrigo A. 1 2
Labra Fabio A. 1 2
Cotoras Darko D. 3 darkocotoras@gmail.com
Camus Patricio A. 4 5
Gutiérrez Dimitri 6
http://orcid.org/0000-0002-0257-2024 Aguirre Luis 7
Rozbaczylo Nicolás 8
http://orcid.org/0000-0001-7736-0969 Poulin Elie 9
Lagos Nelson A. 1 2
http://orcid.org/0000-0001-8235-3990 Zamorano Daniel 2 10
http://orcid.org/0000-0002-1681-416X Rivadeneira Marcelo M. 11 12 13
1 Facultad de Ciencias, Universidad Santo Tomás , Santiago , Chile
2 Centro de Investigación e Innovación para el Cambio Climático (CiiCC), Universidad Santo Tomás , Santiago , Chile
3 Entomology Department, California Academy of Sciences , San Francisco, California , United States
4 Departamento de Ecología, Facultad de Ciencias, Universidad Católica de la Santísima Concepción , Concepción , Chile
5 Centro de Investigación en Biodiversidad y Ambientes Sustentables (CIBAS), Universidad Católica de la Santísima Concepción , Concepción , Chile
6 Dirección de Investigaciones Oceanográficas y de Cambio Climático, Instituto del Mar del Perú (IMARPE) , Callao , Perú
7 Laboratorio de Biología y Sistemática de Invertebrados Marinos (LaBSIM), Facultad de Ciencias Biológicas, Universidad Nacional Mayor de San Marcos , Lima , Perú
8 FAUNAMAR Ltda. Consultorías Medio Ambientales e Investigación Marina , Santiago , Chile
9 Instituto Milenio de Ecología y Biodiversidad (IEB), Facultad de Ciencias, Universidad de Chile , Santiago , Chile
10 Department of Zoology, University of Otago , Dunedin , New Zealand
11 Laboratorio de Paleobiología, Centro de Estudios Avanzados en Zonas Aridas (CEAZA) , Coquimbo , Chile
12 Departamento de Biología Marina, Facultad de Ciencias del Mar, Universidad Católica del Norte , Coquimbo , Chile
13 Departamento de Biología, Universidad de La Serena , La Serena , Chile
Costello Mark
Electronic publication date: 2021 Sep 27
Publication date: 2021
Volume: 9
Electronic Location ID: e12010
Received 2020 May 15; Accepted 2021 Jul 28
Copyright: © 2021 Moreno et al.
Copyright year: 2021
Copyright holder: Moreno et al.
License: This is an open access article distributed under the terms of the Creative Commons Attribution License, which permits unrestricted use, distribution, reproduction and adaptation in any medium and for any purpose provided that it is properly attributed. For attribution, the original author(s), title, publication source (PeerJ) and either DOI or URL of the article must be cited.
License URL: https://creativecommons.org/licenses/by/4.0/

Keywords: Annelida, Biogeography, Macroecology, Macroevolution, Random forest, Niche conservatism

Funding: Rodrigo A Moreno received funding for this study from CONICYT D-21070030 and CONICYT AT-24091021 Instituto de Ecología y Biodiversidad (IEB) Universidad de Chile Servicio Hidrográfico y Oceanográfico de la Armada de Chile and the Comité Oceanográfico Nacional (CONA) CONA-C13F 07-07, CONA-C14F 08-10, CONA-C15F 09-09, CONA-C17F 11-09 and CONA-C18F 12-08 BASAL PFB 023 CONICYT-CHILE and the Iniciativa Científica Milenio ICM P05-002 FONDECYT N° 1100729 Millennium Science Initiative Program (ICN2019_015) SECOS PIA ANID ACT 172037 Darko D Cotoras was supported with his personal funds ANID/FONDECYT 1200843 Concurso de Fortalecimiento al Desarrollo Científico de Centros Regionales 2020-R20F0008-CEAZA Rodrigo A. Moreno received funding for this study from CONICYT D-21070030 and CONICYT AT-24091021 fellowships, Beca de Subsidio a la Investigación for Doctoral Thesis projects from the Instituto de Ecología y Biodiversidad (IEB) and a research grant from the Programa de Estadías Cortas de Investigación of the Vicerrectoría de Asuntos Académicos, Universidad de Chile. Nicolás Rozbaczylo received financial support from CONA-C13F 07-07, CONA-C14F 08-10, CONA-C15F 09-09, CONA-C17F 11-09 and CONA-C18F 12-08 projects through of the Servicio Hidrográfico y Oceanográfico de la Armada de Chile and the Comité Oceanográfico Nacional (CONA). Elie Poulin received financial support from BASAL Grant PFB 023, CONICYT-CHILE and the Iniciativa Científica Milenio Grant ICM P05-002. Fabio A Labra received support from FONDECYT grant N° 1100729. Nelson A Lagos received additional support during the finalization of this study from the Millennium Science Initiative Program (ICN2019_015) SECOS. Fabio A Labra and Nelson A Lagos were supported by PIA ANID ACT 172037. Darko D Cotoras was supported with his personal funds. The research of Marcelo M Rivadeneira was funded by ANID/FONDECYT 1200843 and by the “Concurso de Fortalecimiento al Desarrollo Científico de Centros Regionales 2020-R20F0008-CEAZA". There was no additional external funding received for this study. The funders had no role in study design, data collection and analysis, decision to publish, or preparation of the manuscript.

==============================
Latitudinal diversity gradients (LDG) and their explanatory factors are among the most challenging topics in macroecology and biogeography. Despite of its apparent generality, a growing body of evidence shows that ‘anomalous’ LDG (i.e., inverse or hump-shaped trends) are common among marine organisms along the Southeastern Pacific (SEP) coast. Here, we evaluate the shape of the LDG of marine benthic polychaetes and its underlying causes using a dataset of 643 species inhabiting the continental shelf (<200 m depth), using latitudinal bands with a spatial resolution of 0.5°, along the SEP (3–56° S). The explanatory value of six oceanographic (Sea Surface Temperature (SST), SST range, salinity, salinity range, primary productivity and shelf area), and one macroecological proxy (median latitudinal range of species) were assessed using a random forest model. The taxonomic structure was used to estimate the degree of niche conservatism of predictor variables and to estimate latitudinal trends in phylogenetic diversity, based on three indices (phylogenetic richness (PDSES), mean pairwise distance (MPDSES), and variation of pairwise distances (VPD)). The LDG exhibits a hump-shaped trend, with a maximum peak of species richness at ca. 42° S, declining towards northern and southern areas of SEP. The latitudinal pattern was also evident in local samples controlled by sampling effort. The random forest model had a high accuracy (pseudo-r2 = 0.95) and showed that the LDG could be explained by four variables (median latitudinal range, SST, salinity, and SST range), yet the functional relationship between species richness and these predictors was variable. A significant degree of phylogenetic conservatism was detected for the median latitudinal range and SST. PDSES increased toward the southern region, whereas VPD showed the opposite trend, both statistically significant. MPDSES has the same trend as PDSES, but it is not significant. Our results reinforce the idea that the south Chile fjord area, particularly the Chiloé region, was likely the evolutionary source of new species of marine polychaetes along SEP, creating a hotspot of diversity. Therefore, in the same way as the canonical LDG shows a decline in diversity while moving away from the tropics; on this case the decline occurs while moving away from Chiloé Island. These results, coupled with a strong phylogenetic signal of the main predictor variables suggest that processes operating mainly at evolutionary timescales govern the LDG.

Introduction

Latitudinal diversity gradients (LDG) have been studied for over two centuries and the search for patterns and their processes remain an active topic in biogeography, macroecology and evolution (Pianka, 1966; Willig, Kaufman & Stevens, 2003; Hillebrand, 2004a; Mittelbach et al., 2007; Tittensor et al., 2010; Kinlock et al., 2018; Pontarp et al., 2019). Several meta-analyses have shown the inter-hemispheric consistency of a monotonic increase in species richness from high latitudes to the tropics, producing a unimodal (canonical) pattern on a global scale (i.e., Willig, Kaufman & Stevens, 2003; Hillebrand, 2004a, b; Costello & Chaudhary, 2017; Worm & Tittensor, 2018; Kinlock et al., 2018; Menegotto & Rangel, 2018, 2019; Rivadeneira & Poore, 2020). However, there is considerable debate regarding the underlying factors that determine the LDG (Rohde, 1992; Willig, Kaufman & Stevens, 2003; Mittelbach et al., 2007; Pontarp et al., 2019).

Synthesis achieved during the last decades recognize that explanations for the canonical LDG (i.e., increasing species richness towards the tropics) can be broadly classified into three major hypotheses categories (Mittelbach et al., 2007; Pontarp et al., 2019): ‘ecological limits’, ‘diversification dynamics’ and ‘time for species accumulation’. The category of ‘ecological limits’ refers to explanations based on present-day environmental conditions via higher productivity, carrying capacity, species coexistence, or niche breadth (Currie, 1991; Rosenzweig, 1995; Evans, Warren & Gaston, 2005). The ‘diversification dynamics’ hypotheses suggest that the LDG are generated by latitudinal variation in diversification rates, with greater speciation and/or lower extinction rates in the tropics (Evans, Warren & Gaston, 2005; Weir & Schluter, 2007). The ‘time for species accumulation’ hypotheses explanations invoke the tropics as sources of evolutionary novelties, where clades originate making the LDG the result of dispersal limitation towards subtropical zones (Wiens & Donoghue, 2004; Wiens & Graham, 2005; Jablonski et al., 2013).

In spite of its apparent generality, the canonical LDG pattern is not matched by many continental and marine taxa (see Platnick, 1991; Kindlmann, Schödelbauerová & Dixon, 2007; Tittensor et al. 2010; Worm & Tittensor, 2018, Kinlock et al., 2018). A growing body of evidence shows that non-canonical LDG (i.e., inverse or hump-shaped trends) are common among marine organisms at a global scale (Chaudhary, Saeedi & Costello, 2016; Woolley et al., 2016; Rivadeneira & Poore, 2020). This same phenomenon has been reported along the Southeastern Pacific coast (0–56° S, hereafter SEP) and also involve widely different taxa such as macroalgae (Santelices, 1982; Santelices & Marquet, 1998; Santelices, Bolton & Meneses, 2009), sponges (Desqueyroux & Moyano, 1987), anthozoans (Häussermann & Försterra, 2005), bryozoans (Moyano, 1991), polyplacophorans (Navarrete et al., 2020), gastropod and bivalves (Herm, 1969; Valdovinos, Navarrete & Marquet, 2003), polychaetes (Lancellotti & Vásquez, 2000; Hernández, Moreno & Rozbaczylo, 2005; Moreno et al., 2006) and different groups of crustaceans (Fernández et al., 2009; Rivadeneira et al., 2011). These inverse trends along the SEP have been shown to not be the result of sampling artifacts (see Valdovinos, Navarrete & Marquet, 2003; Rivadeneira et al., 2011).

In this context, the biogeography of benthic polychaetes from the SEP has historically received little attention and available studies have been mainly descriptive (see Lancellotti & Vásquez, 1999, 2000). Indeed, a significant gap in knowledge in the whole Eastern Pacific and Chile in particular, is recognized on a recent global study of polychaete biogeography (Pamungkas, Glasby & Costello, 2021). However, Hernández, Moreno & Rozbaczylo (2005) and Moreno et al. (2006) evaluated the latitudinal change in species richness of benthic polychaetes along the coast of Chile (between Arica at 18° S and Cape Horn at 56° S), finding a hump-shaped LDG with a maximum peak around Chiloé Island (42° S) in southern Chile. These authors proposed that this pattern might be determined by geometric constraints and historical events such as eustatic movements, cyclic effects of El Niño-Southern Oscillation, shallowing of the oxygen minimum zone and glacial advances and retreats, all of which have mainly occurred since the Neogene (i.e., during the last 23 Mya). Although these studies represented a significant advance in understanding the mechanisms that may generate the hump-shaped LDG, their analyses used a low spatial resolution (3° latitude bands) to register species occurrences and constrained the study area to the geopolitical limits of Chile (an arbitrary decision shared by the majority of studies in other taxa. i.e., Desqueyroux & Moyano, 1987; Moyano, 1991; Fernández et al., 2009; Rivadeneira et al., 2011; Lee & Riveros, 2012). In these studies (Hernández, Moreno & Rozbaczylo, 2005; Moreno et al., 2006), therefore, the exclusion of a vast portion of the Warm Temperate Southeastern Pacific biogeographic province (sensu Spalding et al., 2007) could generate spurious results derived from truncating the real biogeographic limits of these polychaete species.

Here, we reappraise the patterns and processes related to the LDG reported for benthic polychaetes in Chile, using a greater spatial extent (3–56° S) and resolution (0.5° latitude) than prior analyses. The northern end is defined by the Guayaquil Gulf (limit between the Panamanian and Peruvian–Chilean Provinces, Vegas-Velez, 1980; Boschi, 2000; Robertson & Cramer, 2009; Ibanez-Erquiaga et al., 2018), while the southern limit consist in the end of the South American continent isolated from Antarctica by the Circumpolar Current. Regarding the southern limit, differences in polychaete community composition have been already reported between the Magellan region and Weddell Sea shelves (Montiel et al., 2005).

The existence of non-canonical LDG along the SEP could be the result of a complex interaction of several factors, instead of a single dominant one. To give explanation to multi-variate and non-linear phenomena a statistical framework which incorporates and differentially weights each factor is required. Our approach would be to simultaneously evaluate the relative contribution of several proxies (SST, salinity, shelf area, etc.) which have been previously associated with specific hypothesis to explain LDG, using machine learning methods. The explanatory hypotheses considered can be separated into three categories: ‘ecological limits’, ‘diversification dynamics’, and ‘time for species accumulation’ (Table 1). These categories have been adapted from the conceptual framework used to study the canonical LDG (Mittelbach et al., 2007; Pontarp et al., 2019).

Table 1 Summary of the main predictions from classic hypotheses to explain LDG.

Hypothesis categories	Specific hypothesis	SST	Salinity	SST range	Median lat. Range	Salinity range	Shelf area	Primary productivity	
Ecological limits	Seasonal coexistence			↓ range,
↑ species		↓ range,
↑ species			
Species-area effect						↑ area,
↑ species		
Energy-dependency							↑ PrimProd,
↑ species	
Diversification dynamics	Long term climate stability				↓ range,
↑ species				
Temperature-dependent speciation	↑ SST,
↑ species							
Time for species accumulation	Niche conservatism	= conditions of origination, ↑ species			
Notes:

The arrows (↓ or ↑) denote a relative increase or decrease on the parameter. The change on the first causes the response on the second.

“Time for species accumulation” does not include the variables “Shelf area” and “Primary productivity”.

Regarding ‘ecological limits’, we will consider three hypotheses: (1) Seasonal coexistence (Valentine & Jablonski, 2015), which predicts that diversity will decline in areas with abiotic seasonal changes (Diversity 1/∝ SST range and Salinity range); (2) Species-area effect (Chown & Gaston, 2000; Valdovinos, Navarrete & Marquet, 2003), which predicts that diversity will be a direct function of the available area (Diversity ∝ Shelf area); and (3) Energy-dependency (Hawkins, Porter & Diniz-Filho, 2003; Evans, Warren & Gaston, 2005), which predicts that diversity will increase with more energy available in the ecosystem (Diversity ∝ Primary Productivity). In the category of ‘diversification dynamics’, we will consider two hypotheses: (1) Long term climate stability (Dynesius & Jansson, 2000), which predicts that more stable areas will accumulate more species (Diversity 1/∝ Median latitudinal range of the species) and (2) Temperature-dependent speciation (Rohde, 1992; Allen et al., 2006), which predicts a positive association between speciation rates and ambient temperature (Diversity ∝ SST). Finally, in the category of ‘time for species accumulation’, we will assess one hypothesis about Niche conservatism (Wiens & Donoghue, 2004), which predicts that species will tend to retain preferences for the ecological conditions where they originally evolved (Diversity ∝ constancy in SST, salinity, SST range, Median latitudinal range of the species, and Salinity range). As all these explanatory variables may reflect processes operating at both ecological and evolutionary timescales, we have selected them as candidate variables to build a machine learning model to explain the polychaete LDG in the SEP.

In addition, the study of phylogenetic diversity may shed light into the role of evolutionary processes shaping patterns of species richness (Davies & Buckley, 2011; Fritz & Rahbek, 2012; Eme et al., 2020). For instance, a species-rich area composed by many closely related species (i.e., low phylogenetic diversity) may imply elevated in situ speciation rates of some lineages. On the contrary, a species-rich area composed by many distantly related species (i.e., high phylogenetic diversity) may be associated to immigration influenced by filtering effect acting upon a trait phylogenetically over dispersed. There are many α-phylogenetic diversity indices (Winter, Devictor & Schweiger, 2013), that can be broadly classified into three categories that measures amount of richness, divergence and regularity (Tucker et al., 2017); therefore, the combined used of indices reflecting each one of these three aspects may be more informative of the role of evolutionary processes shaping species richness (Davies & Buckley, 2011; Fritz & Rahbek, 2012; Eme et al., 2020). Despite the fact that these phylogenetic diversity indices are typically based on calibrated molecular phylogenies, the taxonomic structure may be still be used as a coarse proxy of phylogenetic relationships of species in groups lacking robust and complete molecular phylogenies (Soul & Friedman, 2015). In fact, multiple studies in marine taxa have analyzed latitudinal trends in phylogenetic diversity using this taxonomic approach (Woodd-Walker, Ward & Clarke, 2002; Ellingsen et al., 2005; Tolimieri & Anderson, 2010; Rivadeneira et al., 2011; Azovsky, Garlitska & Chertoprud, 2016; Wu, Chen & Zhang, 2016). This opens the possibility to explore latitudinal patterns of phylogenetic diversity in benthic polychaetes along SEP and its connection with the LDG phenomenon.

Our study area encompasses a broad latitudinal gradient of oceanographic conditions spanning more than 50 degrees of latitude, from tropics to sub-polar areas, and hence it is ideal to study the role of ‘ecological limits’ on the LDG (Astorga et al., 2003; Valdovinos, Navarrete & Marquet, 2003; Fernández et al., 2009). At the same time the marine biota at SEP experienced a drastic turnover during the late Cenozoic, driven by tectonic, physiographic, climatic, and oceanographic shifts (Herm, 1969; Rivadeneira & Marquet, 2007; Kiel & Nielsen, 2010; Villafaña & Rivadeneira, 2014; Rivadeneira & Nielsen, 2017). These alterations induced latitudinal differences in the diversification trends that created the inverse LDG characterizing present-day bivalves and gastropods along the SEP (Kiel & Nielsen, 2010) and may have also driven the LDG of benthic polychaetes (Moreno et al., 2006). Thus, SEP is also an ideal region to study the role of evolutionary and historical processes (i.e., ‘diversification dynamics’, and ‘time for species accumulation’) shaping the LDG. Therefore, the specific goals of our study are: (1) to re-evaluate the existence of a hump-shaped LDG, including the entire Peruvian and Magellanic biogeographic provinces along the SEP, and (2) to jointly evaluate different hypotheses (by significance assessment of proxies), which could explain the formation of a hump-shaped LDG. Our study confirms the existence of a non-canonical LDG, robust to sampling bias, and suggests that the underlying explanations may be linked to processes occurring at evolutionary timescales.

Materials and Methods

Database

This study relies on a database which: (a) includes 643 polychaete species found between 3° S (southern Ecuador) and 56° S (Cape Horn, Chile) inhabiting the continental shelf (i.e., ≤200 m depth) (Fig. 1A), (b) summarizes information compiled from exhaustive reviews of literature (ca. 1,000 bibliographic references), museum collections and field expeditions, integrating more than 160 years of studies over the SEP, (c) was constructed using georeferenced data from geographic information systems (Datum WGS84), (d) was enriched by intensive field samplings conducted over the last decade for updating records and correcting biogeographical biases, particularly in southern Chile where diversity surveys historically had been scarce, specially through the research cruises CIMAR Fiordos 13, 14, 15, 17 and 18 (Comité Oceanográfico Nacional, Chile), and (e) was complemented with existing records from the OBIS database (OBIS, 2021). Taxonomy was cross-validated with the World Register of Marine Species (WoRMS Editorial Board, 2021). Altogether, such information currently constitutes the most complete database of benthic polychaetes in this region of the world. The reported latitudinal ranges of all species used in this study are provided in the Data S1.

Figure 1 Latitudinal diversity gradient of benthic polychaetes along the SEP.

(A) Study area and (B) Regional species richness (using a range-through approach) and local species richness (rarefied species richness, E, using s = 20 individuals per site).

We first determined the maximum and minimum latitude of geographic distribution of each species and assigned species presence into 0.5° latitudinal bands using a range-through approach (i.e., assuming a continuous geographic distribution between their range limits), as commonly done in marine macroecological studies (Roy et al., 1998; Tomašových et al., 2016). OBIS records were used to estimate the latitudinal limits of species beyond our study area along the eastern Pacific, from Antarctica to Alaska. We also examined whether differences in the sampling intensity may potentially mask the overall pattern, that is to say, whether areas of higher species richness may be the result of higher sampling effort. In order to test this potential confounding factor, we used 57 soft-bottom benthic assemblages sampled in the coastal shelf and collated from a literature survey. Since the sampling effort was not the same (i.e., different sampling devices and number of samples) we used the number of individuals reported for each species and each site to carry out a rarefaction analysis. This method estimates the expected species richness for a similar number of individuals sampled (Sanders, 1968; Gotelli & Colwell, 2001). In our case, we set Es = 20 (the site with a smaller number of individuals). The information containing the local occurrences and abundances of 152 species in the 57 local assemblages is presented in the Data S2. We estimated the mean rarefied species richness per 0.5° latitudinal bin and correlated it with the species richness estimated with the range-though approach. In the absence of severe spatial bias in sampling intensity, the correlation between rarefied and regional-scale species richness should be positive and significant. However, patterns of species richness may not be necessarily similar across spatial scales (Gray, 2002; Rivadeneira, Fernández & Navarrete, 2002; Hillebrand, 2004a, 2004b; Edgar et al., 2017), and in fact the relationship between local and regional species richness may be non-linear, i.e., at higher regional species richness local assemblages may be saturated (Ricklefs, 1987; Srivastava, 1999; Rivadeneira, Fernández & Navarrete, 2002). The contrary trend, i.e., a linear relationship between local and regional species richness, suggests that local assemblages may be ‘open’ to the dispersal of species from the regional pool (Ricklefs, 1987). We explored this idea by carrying out a second-order polynomial OLS regression, where the significance of the terms can be used to support the saturation (i.e., significant quadratic component) or unsaturation (significant linear component, but non-significant quadratic term) of local assemblages.

Hypothesis testing

We evaluated the importance of six oceanographic variables (Sea Surface Temperature (SST), SST range, salinity, salinity range, primary productivity and shelf area), which were used as proxies for three hypothesis categories currently used to explain canonic LDG (Mittelbach et al., 2007; Pontarp et al., 2019; Table 1). They were obtained from the Bio-Oracle Database v.2.0 (Assis et al., 2018) and GMED database (http://gmed.auckland.ac.nz) (information provided in the Data S3). Values were averaged over half-degree latitudinal bins, using only pixels on the coastal shelf (<200 m depth). The entire dataset of environmental predictors is available in the Data S3. In addition, we used the median latitudinal range of all species contained at each latitudinal bin as a coarse proxy of the ‘long-term climatic stability hypothesis’. Since many species have latitudinal ranges spanning beyond the study area, we used the actual latitudinal ranges estimated for the entire eastern Pacific coast (from Antarctica to Alaska).

The role of all predictors shaping the LDG was evaluated using a random forest approach. Random forest, a machine learning method, offers multiple advantages over traditional GLM/GLS methods commonly used on previous studies (Kreft & Jetz, 2007; Tittensor et al., 2010): no error structure is assumed, it deals with classification and regression problems, it can handle a large number of predictors, and is robust to overfitting (Breiman, 2001; Liaw & Wiener, 2002). We used standard tuning hyperparameters, setting mtry = p/3 (where p is the number of predictors per tree), and node size = 5. Variable importance was estimated using a conditional random forest which accounts for the possible predictor collinearity (Strobl et al., 2008), based on the method proposed by Altmann et al. (2010), with p-values based on 10,000 permutations, and implemented in the library ranger (Wright & Ziegler, 2017) in R. Nevertheless, the degree of predictor multicollinearity, measured as the variance inflation factor (Dormann et al., 2013; Naimi et al., 2014), was below the threshold of ten commonly used in the literature (Table 2). Partial dependence plots were used to inspect the conditional shape of the predicted species richness vs. selected predictors, using the library pdp (Greenwell, 2017) in R. A partial dependence plot allows us to visualize the functional relationship between the species richness and the predictor variables isolated from the effect of other predictors. The existence of spatial autocorrelation in the residuals of the model, another potential bias on the identification of variable importance (Diniz-Filho, Bini & Hawkins, 2003), was evaluated using a spatial autocorrelogram analysis (i.e., Moran’s I vs. geographic distance) with 1,000 runs in the library ncf in R (Bjornstad, 2019).

Table 2 Environmental predictors of the LDG of benthic polychaetes along the SEP.

		Conditional random forest	Phylogenetic signal	
Proxy	VIF	Conditional importance	p-value	K	p-value	
Median lat. range	6.26	831.525	0.0001	0.307	0.003	
SST	9.33	732.814	0.0001	0.332	0.001	
Salinity	3.71	541.348	0.0001	0.274	0.725	
SST range	3.18	217.978	0.0227	0.290	0.162	
Primary productivity	2.81	149.033	0.0805	0.301	0.007	
Salinity range	1.88	129.715	0.0822	0.286	0.215	
Shelf area	1.64	58.671	0.2966	–	–	
Note:

These predictors are proxies of different hypotheses grouped into three categories (sensu Pontarp et al., 2019). Also shown are their level of collinearity (variance inflation factor, VIF), variable importance according a conditional random forest analysis and their phylogenetic signal (Blomberg’s K). Statistically significant results and their respective p-values are shown in bold.

Although some of the environmental predictors (e.g., SST) may be considered as a proxy for evolutionary hypotheses, we also evaluated the importance of evolutionary processes on the LDG using two additional approaches. We used taxonomy as a coarse proxy of the phylogenetic relationships (Soul & Friedman, 2015), since we lack a full and well-resolved phylogeny for our species. First, we estimated three indices of α-phylogenetic diversity that summarizes different facets of the evolutionary relatedness of species, namely phylogenetic richness (Faith’s PD), divergence (mean pairwise distance, MPD), and regularity (variation of pairwise distances, VPD) (Tucker et al., 2017). PD measures the total distance from each tip to the root of the tree, and it is used as a proxy of the total evolutionary history summed by all species in a given region (Faith, 1992; Winter, Devictor & Schweiger, 2013). The MPD (also known as AvTD, and Δ+, Clarke & Warwick, 1998) measures the mean phylogenetic distance (i.e., branch length) among all pairs of species within a given latitudinal bin. We estimated their standardized effect sizes (SES = [observed − mean simulated]/SD simulated) of PD and MPD in order to account for their dependence on species richness, by randomizing the original species matrix 1,000 times. SES values lower than expected by the null model suggest phylogenetic clustering, whereas values higher than expected indicate phylogenetic overdispersion/convergence. We also estimated the VPD (Warwick & Clarke, 1998) as a measure of the regularity of the phylogenetic distances among all species in each latitudinal bin. Analyses were carried out using the libraries vegan (Oksanen et al., 2019) and picante (Kembel et al., 2010) in R.

As a second approach to estimate the importance of evolutionary processes shaping the LDG, we determined the phylogenetic signal of each proxy, using Blomberg’s K (Blomberg, Garland & Ives, 2003) in order to test the importance of the Niche conservatism hypothesis (within the ‘time for species accumulation’ category). This was carried out using the median value of each proxy estimated across the entire latitudinal distribution. Larger values of K indicate a strong phylogenetic conservatism of the proxy. Analyses were carried out using the libraries ape (Paradis, Claude & Strimmer, 2004), paleotree (Bapst, 2012), and picante (Kembel et al., 2010) in R. The script used to run all analyses and to create each figure is provided in the Scripts S4.

Results

Our compiled data set shows that the species diversity of benthic polychaetes along the SEP presents a hump-shaped LDG (Fig. 1B). The maximum richness (299 species) occurs at 42° S, which is two-fold higher than in southern Ecuador (3° S) where richness was 152 species. A secondary peak is located ca. 53° S, and then species richness drops abruptly towards 55° S, with species richness levels similar to those observed in Perú and southern Ecuador. The evaluation of potential sampling artifacts showed that rarefied species richness in local assemblages has the same trend observed when using the range-through approach (r = 0.49, p = 0.009, n = 29 bins, Fig. 1B), and no evidence of saturation was detected, as the linear but not the quadratic term of the local vs. regional OLS regression was significant (p = 0.008 and 0.179, respectively).

The random forest model showed a high accuracy predicting the LDG (Figs. 2A and 3A), with a pseudo-r2 = 0.95. The model underestimated the species richness in the most extreme latitudinal bins (Fig. 2B), but overall spatial autocorrelation was not detected at any spatial distance (Fig. 2C). Out of the seven predictors, four were significant (median latitudinal range, SST, salinity and SST range; ranked according to its importance, see Table 2). For a reference of the latitudinal variation of the selected variables see Figs. 3B, 3C. Partial dependence shows a hump-shaped response of species richness to SST (Fig. 4A), with maximum values at ca. 12 °C.

Figure 2 Diagnostic plots for the random forest model.

(A) Observed vs. predicted species richness, (B) latitudinal distribution of standardized residuals, and (C) spatial autocorrelogram of the residuals (gray area shows the 95% confidence intervals of a null model).

Figure 3 Environmental predictors of the latitudinal diversity gradient of benthic polychaetes along the SEP.

(A) Observed and predicted species richness by a random forest model, (B) latitudinal variation of SST and SST range, and (C) latitudinal variation of salinity and median latitudinal range of species.

Figure 4 Partial dependence plot showing the relationship between predicted species richness of benthic polychaetes and the selected explanatory variables.

(A) SST, (B) SST range, (C) salinity, and (D) median latitudinal range.

A more pronounced hump-shaped response of species richness was observed for SST range (Fig. 4B). Salinity and median latitudinal range showed a monotonic negative relationship with species richness, which reaches maximum values at the lowest salinity levels (Fig. 4C) and bins with lower median latitudinal ranges (Fig. 4D).

The three phylogenetic diversity indices showed different latitudinal trends and their correlation to species richness varied in sign and magnitude (Fig. 5). While PDSES was positively correlated to species richness (rPearson = 0.51, p < 0.0001, Fig. 5A), MPDSES did not show a significant relationship (rPearson = 0.04, p = 0.69, Fig. 5B), and VPD was negatively correlated with species richness (rPearson = −0.49, p < 0.0001, Fig. 5C). PDSES values were not different than expected by the null models for most of the latitudinal gradient (Figs. 5A, 5B), except by areas around northern Chile and Perú (north of 20° S). Observed MDPSES values were significantly lower than expected in two large areas centered around Chiloé (42° S), and northern Chile and Perú (ca. 15° S). For both PDSES and MDPSES these negative values suggested a phylogenetic clustering pattern. Blomberg’s K was only significant for the median latitudinal range, SST and primary productivity (Table 2).

Figure 5 Latitudinal gradient of phylogenetic diversity of benthic polychaetes along SEP.

(A) Faith’s phylogenetic diversity (PDSES), (B) mean pairwise distance (MDPSES), and (C) variance in pairwise distance (VPD).

Discussion

Our study supports the existence of a hump-shaped LDG for benthic polychaetes along the SEP, increasing the number of exceptions to the canonical pattern of higher species richness towards the tropics, both in the SEP and other regions (i.e., Moyano, 1991; Valdovinos, Navarrete & Marquet, 2003; Willig, Kaufman & Stevens, 2003; Hillebrand, 2004a, 2004b; Kindlmann, Schödelbauerová & Dixon, 2007; Santelices, Bolton & Meneses, 2009; Rivadeneira et al., 2011). Particularly, it is congruent with a recently published global study on latitudinal gradients of polychaetes (Pamungkas, Glasby & Costello, 2021). Our research also confirms previous results for benthic polychaetes along SEP (Hernández, Moreno & Rozbaczylo, 2005; Moreno et al., 2006) based on partial datasets of a more restricted scale and lower spatial resolution (ca., 3° of latitude), and it seems robust to possible sampling artifacts, as the overall shape of the LDG still holds after accounting for differences in sampling in local assemblages. Therefore, the recorded pattern in this study reflects a robust biogeographic pattern for benthic polychaetes in the SEP.

Our analyses reveal that the LDG of polychaete species is well explained by a reduced subset of predictors that could be attributed to processes operating mainly, but not exclusively, at evolutionary timescales (Table 1). Overall, these significant variables (median latitudinal range, SST, salinity, and SST range) are proxies for the predictions of four (Long term climate stability, Niche conservatism, Temperature-dependent speciation, and Seasonal coexistence) of the six ‘specific hypothesis’, corresponding to all three hypotheses categories proposed in previous literature (Mittelbach et al., 2007; Pontarp et al., 2019). It is important to note that all the selected predictors are also linked to ‘time for species accumulation’ category, and particularly to the hypothesis of ‘niche conservatism’ (Wiens & Donoghue, 2004). This hypothesis is independently supported by the detection of significant phylogenetic signal (Blomberg’s K) in the two most important predictors (median latitudinal range and SST). In addition, the LDG is also correlated to spatial patterns of phylogenetic diversity, measured according to two different indices (PDSES and VPD).

In the context of a non-canonical LDG, as our case, the ‘niche conservatism’ hypothesis assumes that clades are originated outside of the tropics. Although we lack paleontological or phylogenetic information to fully test this idea, the prevalence of the median latitudinal range as the most important predictor of the species richness, being also the predictor with a high degree of phylogenetic conservatism, strongly supports the idea of an extra-tropical origin for many polychaetes species along the SEP. Areas of high species richness are associated to latitudinal bins with species with narrower latitudinal ranges, as predicted by the ‘Rapoport’s rule’ (Stevens, 1989; Hernández, Moreno & Rozbaczylo, 2005). Our results reinforce the idea that the Southern Chile fjord area, particularly the Chiloé region, may act as a source of new species of marine polychaetes along SEP creating a hotspot of diversity (Moreno et al., 2006).

A possible historical explanation for the origin of this hotspot is related with the fact that during the Last Glacial Maximum (LGM) the Patagonian Ice Sheet covered a large portion of the fjords probably pushing many species from the canals into the glacial coastal line (Davies et al., 2020). Indeed, it has been shown that the populations of the southern bull-kelp (Durvillaea antarctica) on the fjord area correspond to recent, probably post glacial, re-colonization (Fraser et al., 2010), implying the unavailability of this region during the LGM. Chiloé Island corresponds to the northern most coastal area where the ice sheet reached sea level, indeed the north-west part of the island has remained ice-free since the last 35,000 years (Davies et al., 2020). Therefore, the waters around Chiloé Island, especially in the north-west corner, might have worked as marine coastal refugia preserving species with previous more extended distributions towards the south.

In addition, the relationship between the LDG and phylogenetic diversity indices also suggest the importance of local diversification processes (i.e., origination and extinction, Davies & Buckley, 2011; Fritz & Rahbek, 2012; Eme et al., 2020). Thus, the significant phylogenetic clustering found in the fjords area (i.e., many species are phylogenetically closely related) suggests that this area may have not only acted as a potential glacial refugia, but also a hotspot of in situ diversification, as seen in the fossil record of marine mollusks of the region (Kiel & Nielsen, 2010).

Oceanographically, bidirectional dispersion from this area could be facilitated because precisely at this latitude the West Wind Drift diverges into the Humboldt and Cape Horn currents; running north and south, respectively (Strub et al., 2019). Biologically, the extent of the dispersion might be limited due to the strong niche conservatism detected on latitudinal range and median SST experienced for the species. Additionally, the West Wind Drift might work as a colonization pathway of the SEP from the western Pacific. Altogether, in the same way as the canonical LDG shows a decline in diversity while moving away from the tropics; on this case the decline occurs while moving away from Chiloé Island.

On another hand, the existence of phylogenetic clustering in areas of low species richness such as northern Chile and Perú is more consistent with a negative net diversification trend, produced by high extinction rates coupled to species sorting (via selective survival). Paleontological studies have shown the existence of high extinction rates of marine forms during the late Neogene–Pleistocene along the Peruvian and northern Chile coast (Herm, 1969; Kiel & Nielsen, 2010), which are linked to the Neogene onset of the modern Humboldt Current (Dekens, Ravelo & McCarthy, 2007). In particular, extinction rates of bivalves and marine vertebrates at that time were phylogenetically clustered (Rivadeneira & Marquet, 2007; Villafaña & Rivadeneira, 2014), likely associated to a strong environmental filtering (e.g., thermal tolerance). If similar processes affected polychaetes at SEP, a high extinction rate with higher survivorship of particular clades via environmental filtering process, may lead to an impoverished species richness highly phylogenetically clustered, as it characterizes the northern Chile and Perú regions.

Despite the prevalence of evolutionary-based explanations for the LDG of polychaetes, the lack of phylogenetic signal of the SST range and salinity suggest that processes operating at ecological timescales (i.e., ‘ecological limitations’) may be also important. In fact SST range is much higher in vast areas of Perú and northern Chile, likely related to the effect of ‘El Niño’ events, which governs the inter-annual variability of the sea water temperature (Shaffer et al., 1999). Associated with this and congruent with the selection of SST as a predictor of our model, a recent study also refers to SST as an important factor to globally shape the polychaetes LDG (Pamungkas, Glasby & Costello, 2021).

Conclusions

The SEP provides a natural laboratory for marine biogeography studies given the high diversity of shapes of the LDG in marine taxa. In addition to the existence of canonical LDG (Ojeda, Labra & Muñoz, 2000; Astorga et al., 2003) and inverse LDG (Santelices & Marquet, 1998; Fernández et al., 2009; Rivadeneira et al., 2011), our results validate the existence of a hump-shaped LDG in marine benthic polychaetes centered ca. Chiloé Island (42° S). More importantly, the same conceptual framework used to study the canonical LDG phenomenon could also be used to investigate non-canonical LDGs. The latitudinal patterns of distribution of benthic polychaetes needs to be evaluated in other geographical areas, e.g., central and northeastern Pacific coast, in order to fully understand the generality of the non-canonical LDG, and whether the underlying processes are the same in other regions. In the absence of suitable fossil record for polychaetes, molecular phylogenies of selected clades may shed more direct evidence of the importance of evolutionary processes shaping the LDG in this group. Future studies may be also evaluating the role of dispersal of taxa from other areas of the southern ocean (e.g., Glasby & Alvarez, 1999; Glasby, 2005) shaping the ‘Chiloé hotspot’. An integration of functional traits of species, and phylogenetic and phylogeographic analysis may help to test and validate the importance of ecological and evolutionary determinants of the diversity of benthic polychaetes.

Supplemental Information

Supplemental Information 1 README file for datasets and scripts presented in the article.

Click here for additional data file.

Supplemental Information 2 Database containing the latitudinal distribution and taxonomy of 643 polychaete species along Southeastern Pacific coast.

Click here for additional data file.

Supplemental Information 3 Database containing the local abundance of polychaete species in 57 soft-bottom sites along the Southeastern Pacific.

Click here for additional data file.

Supplemental Information 4 Database containing the environmental predictors used in the article.

Click here for additional data file.

Supplemental Information 5 R-script with instructions to reproduce Table 2 and all figures presented in the article.

Click here for additional data file.

We thank Luis Quipúzcoa, Edgardo Enríquez and Robert Marquina from the Laboratorio de Bentos Marino of the Instituto del Mar del Perú (IMARPE) for logistical help given to the first author in order to consolidate the construction of the benthic polychaetes database from the coast of Perú. The authors dedicate this work to the memory of Prof. Hugo I. Moyano, a Chilean pioneer marine biogeographer and bryozoologist.

Additional Information and Declarations

Competing Interests

Author Contributions

Field Study Permissions

Data Availability

Nicolás Rozbaczylo is employed by FAUNAMAR Ltda.

Rodrigo A. Moreno conceived and designed the experiments, performed the experiments, analyzed the data, prepared figures and/or tables, authored or reviewed drafts of the paper, and approved the final draft.

Fabio A. Labra conceived and designed the experiments, performed the experiments, analyzed the data, prepared figures and/or tables, authored or reviewed drafts of the paper, and approved the final draft.

Darko D. Cotoras analyzed the data, authored or reviewed drafts of the paper, and approved the final draft.

Patricio A. Camus conceived and designed the experiments, analyzed the data, authored or reviewed drafts of the paper, and approved the final draft.

Dimitri Gutiérrez conceived and designed the experiments, performed the experiments, authored or reviewed drafts of the paper, and approved the final draft.

Luis Aguirre conceived and designed the experiments, performed the experiments, prepared figures and/or tables, authored or reviewed drafts of the paper, and approved the final draft.

Nicolás Rozbaczylo conceived and designed the experiments, performed the experiments, authored or reviewed drafts of the paper, and approved the final draft.

Elie Poulin conceived and designed the experiments, performed the experiments, authored or reviewed drafts of the paper, and approved the final draft.

Nelson A. Lagos analyzed the data, authored or reviewed drafts of the paper, and approved the final draft.

Daniel Zamorano analyzed the data, prepared figures and/or tables, authored or reviewed drafts of the paper, and approved the final draft.

Marcelo M. Rivadeneira conceived and designed the experiments, performed the experiments, analyzed the data, prepared figures and/or tables, authored or reviewed drafts of the paper, and approved the final draft.

The following information was supplied relating to field study approvals (i.e., approving body and any reference numbers):

Field collection was supported by research cruises CIMAR Fiordos 13, 14, 15, 17 and 18 (Comité Oceanográfico Nacional, Chile). There was no field study approval number. The research survey was directly supported by the Comité Oceanográfico Nacional, Chile, by allowing researchers to stay on board the research cruises CIMAR.

The following information was supplied regarding data availability:

Raw data and and R script are available in the Supplemental Files.

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
