# Peer review of "Evolutionary drivers of the hump-shaped latitudinal gradient of benthic polychaete species richness along the Southeastern Pacific coast"

_PeerJ, doi:10.7717/peerj.12010_

## Round 0.1 · original submission · Major Revisions

We received three very thorough and constructive referees reports, with extensive recommendations for improvements. I agree with their comments and find the MS needs major revision and further review to be considered for publication.

Reviewer 1 ·

Basic reporting

no comment

Experimental design

no comment

Validity of the findings

no comment

Additional comments

In this paper the authors aim to contrast different ecological hypothesis underlying a latitudinal pattern of marine benthic polychaetes diversity. The authors argue that they are testing five ecological hypothesis: species energy, species-area, Rapopport`s rule, habitat heterogeneity and geometric constraints. In addition the authors argue they are testing two evolutionary hypothesis “diversification rates” and biogeographic conservatism. I recognize the importance and relevance of this study and I believe it has potential, but I believe that at this point, the manuscript is not ready for publication. A series of improvements and modifications are necessary before the manuscript is ready for publication.

## Major points:.

Data and code availability: Before diving into the major points, I would like to point out that the authors decided to submit their paper to a journal that has a very clear polity about data sharing. In their statement, the main author affirm : "I am not submitting raw data or code - Raw data can be found on referenced literature and museum catalogs”. I understand that many of us still have an ancient urge of being the keepers of our own data, or "great collectors”, but that’s not how science advances. I would like to point out that what guarantees that this paper is reproducible is the availability of data and code. In addition, making the data available also guarantees that any researcher trying to investigate these questions in a broader taxonomical context could use the data and even fill many sampling bias for this region without the unnecessary visits to museum catalogs and dozens of papers. Please, make sure that any data and the code used for the analysis are available as supplementary material.

Analysis - Here the authors used OLS and contrast models with Akaike criteria. Although citing Burnham and Anderson’s book, I’m not sure the authors are indeed familiar with multi-model inference as there are two main problem here: a) this inferential framework do not fit the aim of this study and b) the authors use significance tests and model selection at the same time, which comes from two different inferential frameworks that should not be mixed. Model selection is designed for predictive purposes. The best model is the one that maximizes the expected predictive accuracy (i.e. the ability to fit future data). Because the authors are not searching for a model to perform predictions, this inferential framework is wrongly applied here. In addition, even if that was the case, the authors only present the AIC weights in their results, which alone does not mean anything. The authors also tested the significance of each OLS and inform significant models, but model selection and hypothesis testing are different frameworks of statistical analysis with completely different aims and should never be mixed, as pointed out by the book they are citing. To solve all of this problems, I suggest removing model selection from the study and maintaining only the frequentist portion of the analysis - OLS with significance tests. However, in order to do so, the authors should not only exclude model selection from their study but also improve how they are presenting the results. For each OLS it is necessary to present all the statistics for all variables that were used in each model, including the eigenvectors that were used to remove spatial autocorrelation from models residuals. In addition to presenting all the statistics, it is also important to present, as supplementary material, the Moran’s I correlogram in order to evaluate if the spatial autocorrelation on model’s residuals were indeed removed. Also, it is important to be aware about all the assumptions of linear models used here and perform any necessary transformation in predictor variables.

Hypothesis are not independent - The authors separated their hypothesis and even variables that are part of the same hypothesis in different linear models. But, why should energy hypothesis be analysed independent from spatial heterogeneity? In fact it sounds strange to have several independent linear models when only one model with several variables can be easily defined. In fact, when doing so the authors will have access to all statistics and significance test for all variables. Thus, I suggest analyzing as one OLS and discussing those variables that show significant effect. In this case, please make sure to use standardized coefficients in order to compare which variables strongly affects the LGD and which do not. In the current version, it is not possible to compare the different variables because they are tested in different OLS and contrasted with model selection, which informs the best model for prediction. There is one additional problem here regarding the different hypothesis. The authors are testing at the same time hypothesis that were formulated to explain the LDG, a biogeographical effect (Rapoport’s rule) and a null model (Mid domain effect) and contrasting them in order to find the best hypothesis. Rapopports’s effect is a pattern, not an explanation for LDG. Thus, it should not be compared with other hypothesis that propose explanations. In addition the goodness-of-fit of other explanatory hypothesis should be compared with the goodness-of-fit from the mid-domain model. The mid-domain effect is a null model for LDG, not a causal explanation.

Evolutionary hypothesis - I’m not sure how robust it is to infer diversification rates and ‘conservatism’ from taxonomic relationships. I am supposing that phylogenetic information for the studied group is not available. Is that indeed the case? If so, I suggest not using the term diversification, but only using what the authors are indeed calculating - taxonomic distinctness. When using the term biogeographic conservatism, I suggest using biogeographic taxonomic signal. As the authors are aware, what they do when they compute the Moran’s I is to test a signal of a trait in the taxonomy. It is not possible to safely infer 'conservatism' from these signals because there is no evolutionary model being contrasted with the observed data. In addition the authors should clarify if they are testing the Moran’s I only for the first class of distance of the taxonomy, or for the entire taxonomic distance matrix. Finally, the values of Moran’s I observed are very low, which might not be sufficient to even consider that a positive signal exists.

Finally, as I detail in some of my minor points, the authors should revise their introduction pointing out to recent updates in the LDG literature. It is very important to change how they classify the different hypothesis in the introduction and solve some minor problems in few specific paragraphs that are not consistent with recent literature.

#Minor points

Line 62 - 64 - “Latitudinal gradients of species richness (LGRs) and their explaining factors are key topics in macroecology and biogeography, providing a critical framework for ecology, evolution and conservation” - What exactly provides a critical framework for ecology, evolution and conservation ? LGD and their explaining factors ? I am not sure about that. I recommend being more precise here. Maybe saying that that LGD is the oldest pattern in ecology and that understanding its underlying causes is still one the most challenging tasks in ecology and evolution.

Lines 70-85 - More recent classifications exists and should be used here to avoid any confusion (see Pontarp et al. 2019 - The latitudinal Diversity Gradient: Novel Understanding through Mechanistic Eco-evolutionary Models - Trends in Ecology and Evolution). Note that the energy hypothesis is not, in any sense, separated from evolutionary hypothesis, as energy itself might affect diversification rates. In addition, the authors should consider revising hypothesis for marine biodiversity (e.g. the book from Worm and Tittensor - A theory of Global Biodiversity).

Lines 95 - 97 - “Recent works indicate that these inverse trends along the SEP are robust to sampling artifacts and are not the product of statistical artifacts due to a reduced spatial scale (see Valdovinos et al. 2003, Rivadeneira et al. 2011)”. - The most recent study cited here is from nine years ago...Recent studies, at global scale, shows exactly the contrary from what has been affirmed here (see Menegotto & Rangel 2018 - Mapping knowledge gaps in marine diversity reveals a latitudinal gradient of missing species richness - Nature communications).

Lines 100 - 102 - “...as well as the geometric
constraint-mid domain effect hypothesis (MDE hereafter) (if species ranges are random, they will tend to accumulate towards the middle of a bounded domain)...” - Species range are not random in MDE. Please clarify.

Lines 110 - 112 - “However, recently Hernández et al. (2005) and Moreno et al. (2006) evaluated the latitudinal change in species richness of benthic polychaetes along the coast of Chile” - Are these the most recent studies? Although the authors use recently, it has been at least fifteen years since that publication.

Lines 118-126 - Although the authors criticize studies conducted in other resolution, using one single finer resolution does not solve the problem criticized here as we know many ecological processes and patterns are scale dependent. Did the authors test whether the results they observe are consistent across different resolutions? If not, why not ? In addition, why would 0.5 be the best resolution for the studies group?

Lines 136 - 137 - “Filling a gap in knowledge, which will contribute to put in perspective the generality of well-known macroecological hypotheses”. As I pointed out before there is one biogeographical pattern mixtured with ecological hypothesis. In addition, I suggest changing the last phrase and discuss the novelty of this study. In theory, all macroecological studies does what the authors affirm here, thus, what is the novelty about their study?

Line 255 - “tbiogeographic” - Correct typo.

Line 327 - “ might could” - correct phrase.

Reviewer 2 ·

Basic reporting

In this manuscript the authors revisited the latitudinal diversity gradient (LDG) of polychaetes in the southeastern Pacific coast, reanalyzing the diversity pattern described in previous studies and testing a set of underlying mechanisms that could explain it. The text is, overall, well written. I have only minor suggestions to clarify some sentences, which I highlighted in the minor comments below. Literature is well referenced, especially regarding previous studies in the study area. However, it seems that many relevant and recent discussions about the marine LDG are missing. The most recent citation about the marine LDG was published in 2015. Many papers have been published since then, using massive datasets, showing groups that follow the general LDG expectation, groups that have a peak at mid-latitudes, differences regarding the scale of analysis, the limitation of sample bias, evolutionary discussions, etc. Some critiques and suggestion below could be avoided by a revision of the most recent literature. Figures are relevant and clear. Raw data was not supplied (or at least I could not find it).

Experimental design

The research question is well defined, reinforcing previous ecological patterns and providing new hypotheses to explain the sometimes inverse LDG observed in the southeastern Pacific coast. I think that some methods need some clarification, which I highlighted in the minor comments below. The authors carefully considered the sampling bias limitation. One solution adopted here was to use the rarefaction techniques, that have been constantly used in marine studies to account for spatial variations in sampling effort. However, it is interesting to note that the number of benthic assemblages investigated (i.e. re-sampled) also increases toward high latitudes (Fig. 2). I am wondering how much is this efficient to investigate spatial variation in sampling bias. Would it be possible to apply the estimation using the same number of samples or individuals by latitudinal band? Still related to this topic, I am not sure if I agree with the solution of using the year in which the species was described as a measure of robustness to sampling artefact. I understand that this solution seems appropriated to estimate the inventory completeness of the whole study area, but I am not convinced about its efficiency regarding the latitudinal variation in sampling effort. Assuming a hypothetical scenario of historical undersampling at lower latitudes, couldn’t this method simply reflect the historical sampling bias through time (since most of the species would not be recorded or even scientifically described at these regions)? It would be interesting to hear the opinion of the authors about these concerns.

Validity of the findings

I think that some conclusions are contradictory, or at least not so robust. First, it does not make sense to conclude that the mid-domain effect is driven by evolutionary process (lines 317-319). The mid-domain effect provides a null hypothesis in macroecology, where species’ range is distributed at random. Therefore, the results provide both support for a null hypothesis and support for the historical/evolutionary hypothesis. In this case, can we assume an evolutionary process when the resulting pattern does not differ from the null expectation? I think that it is important to think about this question, especially when the evolutionary process identified here were not properly based on phylogenetic analyses, and the taxonomic distinctiveness results does not seem so robust (only one or two bands showed significant results and were highly close to the confidence interval; Fig. 6).
Regarding the biogeographic conservatism result (that species in the Northern limit present more taxonomic conservatism than those at the Southern limit), I am wondering if the simple migration process couldn’t also generate the same result instead of latitudinal differences in diversification or evolutionary restrictions (lines 337-340). Considering that the biota at the Southern part of the study area is closest to the Atlantic ocean, lineages with a completely different evolutionary history could have migrated to the Pacific coast and colonized the Southern limit (I am not sure, just wondering here). It would be interesting to hear from the authors of this scenario is possible and if it could affect the interpretation of these results.
Lastly, if I am not mistaken, spatial autocorrelation should affect the p-value of the regression, not the r-squared. However, all r-squared that you present after including spatial filters in the model are strangely high. Are these r-squared referent to the full model (predictor + spatial filters)?

Additional comments

Minor comments:

Lines 65-67: The statement is correct, but the references are not totally accurate. The sentence informs about “several meta-analyses” while only one actual meta-analysis is cited. I suggest either change the term “meta-analyses” here or to cite proper meta-analyses to be consistent. There are recent publications revisiting this topic in both terrestrial and marine realms.

Line 113: I suggest to include the latitude of the Chiloé Island to facilitate the comprehension of the described pattern.

Lines 136-137: Please, check this sentence and rewrite if necessary.

Lines 146-148: Mixing literature/museum records with field work to fill historical knowledge gaps is a very constructive practice. Well done!

Lines 162-164: I am not sure if I understood your rarefaction method. What are these 71 benthic assemblages? Are they specific locals identified in your dataset? How they are distributed? And why re-sampling only 15 individuals in your dataset? Do you have information in the individual level (maybe only in these 71 assemblages)? Please, clarify.

Lines 165-166: Note that the shape of the LDG may change drastically regarding local and regional scales (see this study of Edgar et al., 2011 [doi 10.1126/sciadv.1700419] and the meta-analyses of Hillebrand).

Line 192: Since you are dealing with benthic species, would not be more appropriated to use sea bottom temperature instead of sea surface temperature?

Lines 205-206: I am not sure of I understood. Did you run a mid-domain model previously to set the number of species in another mid-domain (what I think that does not make sense), or did you run a mid-domain model using the median number of expected species in the whole study area? Please, clarify.

Lines 215-217: I liked the methodological rigor you had here. Well done!

Line 220: Moran Eigenvector Maps (MEM) approach usually return the MEMs where the spatial autocorrelation was statistically identified. Why use the first 10 instead of using only those that may be, in fact, necessary?

Lines 224-226: Have you considered to conduct a multiple regression analysis, considering then the interaction of the predictor variables instead of using individual OLS regressions (i.e. reducing the origin of the LDG to only one single explanation)?

Line 237: “Values below delta+ indicate…”. Do you mean “low values of delta+ indicate”? Since delta+ is an index, it does not make sense to describe a value below the index, right? Please, verify and correct if necessary.

Line 255: Replace “tbiogeographic” by “biogeographic”.

Lines 377-378: I did not understand the importance of nutrient contribution here since your results suggest that primary productivity is not a predictor of species richness.

Line 381: “...there could be global effects that leave…”. What global effects? Please, be more specific here.

Lines 509-510: The link provided in this reference is not working.

Reviewer 3 ·

Basic reporting

The manuscript "Mid-domain effect, biogeographic conservatism and in situ diversification explain the hump-shaped latitudinal gradient of benthic polychaete species richness along the Southeastern Pacific coast," shows new and relevant results for annelids and are written in a rational hypothesis-drive context. The quality of the figures and tables are excellent.
I am positive about the manuscript, but it needs a careful upgrade in the background/context. The literature about the Latitudinal diversity gradient (LDG*) for marine species needs to be updated.
There is a good theoretical introduction about mid-domain effect, LDG, biogeography, ecological, historical and evolutionary hypothesis, but needs updated references about the LDG debate as (among several others):
- Chaudhary et al. (2016, 2017, Trends Ecol & Evol)
- Costello et al. (2017, Nat Commun 8, 1057)
- Kinlock et al. (2018, Global Ecol & Evol 27, 125-141)
- Menegotto & Rangel (2018, Nat Commun 9, 4713)
- Menegotto et al. (2019, Global Ecol & Evol 28(11), 1712-1717)

*Latitudinal diversity gradient (LDG) has been used more often in the literature than Latitudinal gradients of species richness (LGRs)

Experimental design

Two major concerns in the experimental design:
1- The first goal is (1) to re-evaluate the existence of a hump-shaped LGR (or LDG), as know as the constraint-mid domain effect hypothesis (MDE). There are enormous debates about this hypothesis, analytical methods, and explanations. However, I am skeptical about the power of the MS's analyses to address such a hypothesis with just one tail (Southeastern Pacific) of the hump-shaped curve and one taxonomic group (annelids). Also, recent studies have proposed that the latitudinal gradient
of species richness in the marine environment is bimodal (See Chaudhary, C., Saeedi, H. & Costello, M. J. Bimodality of latitudinal gradients in marine species richness. Trends Ecol. Evol. 31, 670–676, 2016; Woolley, S. N. C. et al. Deep-sea diversity patterns are shaped by energy availability. Nature 533, 393–396, 2016; among others cited above). It should be considered in the context of the first hypothesis.
2- Indeed, the taxonomic distinctness index seems an excellent approach as a surrogate of phylogenetic diversity, but the Linnaean classification tree is not an easy task for Annelids. First, since Polychaeta is a paraphyletic class of Annelida, avoid using this terminology (polychaetes) in an ms trying to elucidate the latitudinal gradient of this group. It is common this terminology in an informal conversation with colleagues, without any restriction, but scientifically is recommend to avoiding the use of polychaete/polychaetes/Polychaeta. Instead, use annelids or marine annelids. Also, I could not track precisely which Linnaean classification for annelids was used. A validation matching your species name data in the World Register of Marine Species (WORMs) is essential for using the same Linnaean classification, which is trackable and reproducible.

Validity of the findings

The conclusions are well stated and there impact and novelty, but a literature update in the introduction and the first hypothesis is essential before to review the findings carefull

---

## Round 0.2 · Major Revisions

I have three referees' reports. One has significant concerns about the statistical analysis, a second provides many points of concern and questions the interpretation, and the third a minor criticism. I must add that I agree that it is hard to use such a limited latitudinal gradient to infer evolutionary drivers. At this spatial scale it is likely that regional and local factors dominate species diversity. I think it wise to more critically review the aims and interpretation of this study.

Reviewer 1 ·

Basic reporting

no comment

Experimental design

no comment

Validity of the findings

no comment

Additional comments

I am happy to have the chance to review this manuscript again. I was ‘reviewer #1’ from the first round of reviews and it is a pleasure to observe how this study improved in quality and clarity. Before the manuscript is ready for final publication I have two additional methodological comments regarding how the authors dealt with spatial autocorrelation and collinearity in their modeling approach.

--Spatial autocorrelation
The authors present a Moran’s I correlogram which shows a Moran’s I of ~0.20 in the first class of distance (Fig 2). When dealing with spatial or phylogenetic autocorrelation this amount of autocorrelation is a considerable autocorrelation structure in model’s residuals that should be removed if the author's desire to discuss the significance of their predictors effects, which seems to be the case here. Thus, because there is spatial autocorrelation in the first class of distance, the replicates are not independent and the degrees of freedom are lower than what is considered when computing the significance of the predictors. Thus, calculations of p-values might be biased. Usually, when Random Rorest (used in this study) is applied in ecology the buffer distances from all variables records are included as covariates in the model which ‘teaches’ the model the first law of geography: "Everything is related to everything else. But near things are more related than distant things." These covariates act as statistical ‘band Aids’ that deal with the spatial autocorrelation structure in the residuals. The authors might want to do something similar or just include PCNM axis, built with a distance matrix, as covariates of their model.

--Model collinearity
I understand that it is a cultural aspect of science to use “what other people did in their papers'' and that is one of the reasons why VIF thresholds >10 are still used as an indicator of multicollinearity. However, over the decades, there have been intense debates in statistics about this threshold which has been shown as a very problematic threshold allowing the inclusion of serious problems of collinearity in model building. Below the authors will find references that show that VIF > 5 are a cause of concern and >10 indicates serious collinearity problems. Some more conservative, which I am particularly more familiar with, use a threshold of 2.5. I suspect the authors are using a threshold of 10 because SST has a VIF of 8.5, which is red flag of the model in my opinion. Thus, my suggestion here is to remove SST from the model.

Menard S. Applied Logistic Regression Analysis. 2nd edition. SAGE Publications, Inc; 2001.

James G, Witten D, Hastie T, Tibshirani R. An Introduction to Statistical Learning: With Applications in R. 1st ed. 2013, Corr. 7th printing 2017 edition. Springer; 2013.

Johnston R, Jones K, Manley D. Confounding and collinearity in regression analysis: a cautionary tale and an alternative procedure, illustrated by studies of British voting behaviour. Qual Quant. 2018;52(4):1957-1976. doi:10.1007/s11135-017-0584-6

I would like to congratulate the authors for their intense work in this revision and I wish you all the best.

Reviewer 2 ·

Basic reporting

no comment

Experimental design

no comment

Validity of the findings

no comment

Additional comments

In this new version of the manuscript the authors really improved the data analysis and addressed satisfactorily all the questions raised in my previous review. However, verifying the dataset used to conduct the analyses (only provided in this round) and the interpretation of the new results, I see two major red flags that need to be solved by the authors before publication. These two red flags, which are both methodological and conceptual, reduce the consistency of the results and, more important, the conclusion about the main drivers of the LDG in the SEP. Below I explain in detail all my major and minor points to the authors.


MAJOR POINTS:
If I understand it right, here you use the same hypotheses used for explaining the canonical LDG. In the text you state the you relax some assumptions of the hypotheses (L139), but which assumptions of which hypotheses specifically? They seem all the same to me. Using the same hypotheses is not a problem (I actually think this is more appropriated), what I do not understand is why you do not assume previously that some hypotheses are better than others. For example, why should speciation- temperature relationship explain the higher species richness at 42°S when the SST increases toward the equator? If you are revisiting the same hypotheses used to explain the canonical LDG, you could list the main expected explanations for this situation in which the hypotheses were not proposed. Moreover, you should also discuss these hypotheses in the context of your results (see details in the last major comment).

You affirm that you have a dataset with 600 species (L189) and that the taxonomy was cross-validated using WoRMS (L198). However, checking the Dataset S1 I saw that many species still have unaccepted names (e.g. Ampharete elongata) or more than one entry point due to synonyms (e.g. Hemipodia simplex x Hemipodus simplex), misspelling names (e.g. Aglaophamus heteroserrata x Aglaophamus heterosecirrata) and subspecies (e.g. Abarenicola affinis x Abarenicola affinis chiliensis). According to the R script provided, you used this dataset without taxonomy correction to conduct the analyses. The problem here is that this absence of taxonomic correction may affect the number of species and, more important, the estimated range of the species. See, for example, the latitudinal range of Abarenicola affinis (-9.23 to -13.29) and Abarenicola affinis chiliensis (-32.58 to -33.45) or Aglaophamus polyphara (-25.42 only) and Aglaophamus polypharus (-14.23 to -15.34). I strongly recommend correcting the scientific names, merge the records when necessary and repeat the analyses to guarantee more reliable results. Check also the taxonomy of the species in the Dataset S2.

The first red flag. I understand the decision of excluding from the range size calculation those species occurring at the boundary of the SEP because your data search was focused on this biogeographic region and, in this case, the latitudinal range of these species would remain underestimated (please, feel free to correct me if my point is not correct). However, checking now the Dataset S1 I realize that you could have verified the latitudinal distribution of the species based on records from public global databases such as OBIS and GBIF. I think that this is important not only for those species at the boundary of the SEP but especially for the other species because their latitudinal range can be strongly underestimated. By quickly checking the Dataset S1 and searching for the distribution of the species in global databases we can find species that in your dataset are considered restricted to the SEP but that actually extends to the northern hemisphere (e.g. Alitta succinea, Acromegalomma pigmentum), to the Antarctica (e.g. Aglaophamus trissophyllus, Amage sculpta), or just have a distribution in the SEP wider than you found (e.g. Aricidea pigmentata, Boccardia tricuspa). Despite the impressive effort to assemble species records from the entire SEP, ignoring the occurrence of these species out the SEP region may be affecting your richness estimate and, most probably, the estimated range size of the species. The exclusion or underestimation of range size for species in the southern latitudes decreases the median range size at this region, which may bias some results. In addition the underestimated range will also affect the analyses of niche conservatism.

Please, check carefully the Fig. 3, Fig. 4 and Fig. 5 because some distributions are inverted. For example, in Fig. 3 the plot “b” seems ok (SST) but species richness is peaking in lower latitudes (plot “a”) and median range size is peaking in the higher latitudes (plot “c”), contrary to the description. I added other questions about the Fig. 4 in the minor comments below. Check also if this latitudinal inversion occurs only in the figure or if it is affecting the analyses.

The second red flag. The results do not seem robust to support the conclusion. You claim that evolutionary processes drive the hump-shaped LDG in the SEP based on the importance of three environmental variables, median range size, and results from phylogenetic diversity. First, the important environmental variables to the model were SST, Salinity and SST range. Here, only SST is a proxy of diversification dynamics. However, the relationship between SST and species richness is quadratic (Fig. 4A, peaking at 14° C), contrary to the theoretical assumption regarding the effects of temperature on increasing speciation rates. Note that you use the same hypotheses used to explain the canonical LDG (Table 2), but do not create previous expectations based on the already known hump-shaped diversity distribution in the SEP. For example, why should temperature explain a higher diversity in the mid-latitudes according to the temperature-dependence speciation hypothesis? This is not clear in the introduction and it is completely ignored in the discussion. Salinity (average) does not even appear in the Table 2 or in the introduction (L142-159), it is only used to evaluate niche conservatism. So we (the readers) were not informed about what would be the expected effects of salinity on increasing species richness and how it supports the conclusion. SST range is an hypothesis related to the “ecological limits” category, not “diversification dynamics”. Second, the results of the median range size may change since many species have their latitudinal limits underestimated (as explained above). The increasing range size in the southern region of the SEP (well possible since many species seem to be distributed on Antarctic waters) can reduce the median range size variation and maybe the importance of this predictor. Finally, the phylogenetic diversity indices have contrasting interpretations across the text. In the results section it is shown that PD and MPD have values lower than expected from the null model most in the north (lower latitudes, though the Fig. 5 seems inverted), indicating reduced phylogenetic distance among the species (L314-315). In the discussion section, the low values of MPD are used to reinforce the idea of high speciation rates in the mid latitudes (L352-354), contrasting the results. Although there are some low values around 40°S, why is not the result of the MPD indicating higher speciation rates in the lower latitudes (according to the assumptions present in the text)? Why the low values of MPD in mid-latitudes would indicate speciation while the low values in lower latitudes would indicate species sorting (L379-380)? This change of interpretation is really confusing. Moreover, automatically assuming that low values of PD and MPD directly indicate higher speciation rates is seriously misleading since they may also indicate lower speciation rates in different contexts (see Eme et al. 2020. Ecography, 43, 689-702). Altogether, I cannot see robust evidences of high speciation rates in the Chiloé region explaining the hump-shaped LDG along the SEP as the study proposes.


MINOR POINTS:

L42: I think that the acronym should be consistent with the original expression. I agree with the other reviewers that “latitudinal diversity gradient” is more common and, therefore, LDG may be used, but I do not understand why using LDG after latitudinal gradients of species richness. Why do not use “latitudinal diversity gradient” and be consistent with the LDG acronym? Same for line 73.

L83: Plural sentence?

L106-108: I think this sentence needs some clarification. The inverse trend is not a product of statistical artifacts because the spatial scale is reduced? Which scale are you talking about here? Please, clarify.

L111: I suggest remove “However” here because, if I got this right, this sentence is not contrasting your previous statements. In the previous paragraph you affirm that many taxa have non-canonical LDG and now you are detailing the known pattern for polychaetes, which is also non-canonical.

L118-120: I did not understand if these studies tested some of the hypotheses and their main limitation is only the reduced spatial resolution. If they tested some mechanistic hypotheses so these studies are not descriptive as you stated before (L109-110).

L127-129: One-sentence paragraph?

L133: Do you mean “One approach”?

L150-151: I suggest being clearer here. Many readers may not be aware about how range size may be related to species accumulation. You could be more explicit about the mechanism linking the proxy and the expected result.

L159: Which tuning? Did you do that? This sentence seems out of context to me. Please, clarify.

L217: Do you mean “range-through approach”?

L238-239: Please, check the grammar of this sentence.

L256-257: You consider here that the investigated relationships may be monotonic or non-linear, but a monotonic relationships already include both linear and non-linear distributions, right? I suggest clarify these expected relationships (e.g. quadratic, etc) to avoid misinterpretation.

L286/288: Here you use “traits” referring to the investigated environmental predictors. Is that really appropriated?

L299: Please, check if this values are correct. I found some values slightly different running the script (e.g. n = 28).

L306: Here you call the Fig. 2b and 2c to show results regarding the effect of the predictors in the species richness, but I did not understand what we should be seeing here since these figures only show the residual and the Moran’s correlogram. Please, clarify.

L308: Peak at ca. 12-14° C. In the Fig. 4a the peak seems to occur after the 15° C. Is this x axis also inverted?

L309-310: Are you sure that the Fig. 4b (SST range) shows a hump-shaped relationship?

L310-311: Are you sure that the Fig. 4c (Salinity) shows a negative relationship?

L313-315: This description apparently includes both Fig. 5a and 5b, but the representation of which values were different from the null model or not is only available to Fig. 5b. Why the p-values of the Fig. 5a are not available?

L316-317: Please, check the grammar of this sentence.

L319: Do you mean -0.81 (negative relationship)?

L322: What about the phylogenetic signal in those variables that do not reach the statistical significance? What that means?

L365: Check the grammar in “area are where”.

L383: The influence of El Niño and the bidirectional currents (371) on species distribution along the SEP are very interesting ideas.

L711: It is difficult to read a dataset with comma-separated values when one variable also have comma inside (author, year). I recommend using another separator.

Reviewer 3 ·

Basic reporting

The ms has a significant improvement. The authors addressed all the comments made by our revision and can be acceptable in this current format.

Experimental design

The research questions are more suitable in its current state and fit the reproducibility criterium

Validity of the findings

There are several intersting novelts, rational, and with the literature clear stated. It is a bit sad the authors kept their argumentation to use the term "polychaete." They are entirely right in their argumentation/answer. Polychaeta is still a valid taxon in WORMs, but it is a clear example when Linnean nomenclature rules are ambiguous with the phylogenetic hypothesis. This priority nomenclature rule makes these tasks even more complicated. Anyway, we will ensure survival; among the steps that science needs to advance in our society, this one is a minor concern.

---

## Round 0.3 · Minor Revisions

Both referees compliment you on the improvements to the paper. Referee 2 makes some good technical suggestions that will tighten up the paper and referee 1 asks one question that could be addressed in the paper. With those tasks done your paper should be ready for publication.

Reviewer 1 ·

Basic reporting

No comment

Experimental design

No comment

Validity of the findings

No comment

Additional comments

This is the third time I am reviewing this paper. It is a pleasure to see how the paper evolved in quality since the first round of revisions. I have no more major comments for this manuscript and I think the authors did a good job on responding and incorporating the reviewer’s comments. Thus, I think this paper is ready for it’s final publication. I would like to congratulate the authors for their work and I wish you all the best!

#Minor comment

I understood the authors points about the high VIF values but I think it is important to explain in the results or in the results table why high values of VIF are not a problem given the methods used. It is something that calls the attention of anyone who will read this paper and it might sound like a major error in methodology if the reader skips directly to the tables.

Reviewer 2 ·

Basic reporting

no comment

Experimental design

no comment

Validity of the findings

no comment

Additional comments

I am glad to see how this manuscript have been improved. The authors made a major work reinforcing the data quality, data analysis and text clarity. The study seems much more robust now and I believe it brings a solid contribution to the marine ecology literature. I am satisfied with basically all modifications and responses provided by the authors.

There is only one last commentary that I believe was not totally solved at this round. The authors informed that all scientific names have been checked on WoRMS. However, there is still a few subspecies in the dataset. This means data some species are counted twice in a latitudinal bin (e.g. Maldane sarsi), or their latitudinal extent is splitted in different bins (e.g. Capitella capitata). Since the study was conducted at the species level, these subspecies should be renamed and the latitudinal extent of the species should be revised. I sincerely believe that this correction won’t change the results, neither will need major changes in the text, but I strongly suggest dedicate one more day of revision to correct the dataset and re-run the analyses, just for conscience’ sake.

To accelerate the process, I took the liberty of checking the names using an algorithm to inform the authors all species/subspecies that need some review. Here are the subspecies, and their accepted species name: Capitella capitata capitata [Capitella capitata], Dodecaceria laddi oculata [Dodecaceria laddi], Euphrosine cirrata magellanica [Euphrosine magellanica], Lepidonotus crosslandi peruana [Lepidonotus crosslandi], Maldane sarsi antarctica [Maldane sarsi], Melinna cristata australis [Melinna cristata], Neoamphitrite affinis antarctica [Neoamphitrite affinis], Nereis pelagica lunulata [Nereis pelagica], Notoproctus oculatus antarcticus [Notoproctus oculatus], Exogone obtusa tasmanica [Parexogone minuscula], Sphaerosyllis capensis chilensis [Sphaerosyllis capensis], Terebella lapidaria juanensis [Terebella lapidaria]. This species name is incorrect: Exogone lourei [Exogone (Exogone) lourei]. This species name is invalid and should probably be removed from the dataset: Mediomastus branchiferus [taxon inquirendum]. Check also the dataset 2, some of the names are also present there.


Minor comments:

L44: 651 species. This number must be revised because there are subspecies in the dataset. Same to L203, L744.

L261: “used on previous in studies”. Revise sentence.

L269: “showed a variance inflation factor”. Do you mean a “high variance inflation factor”? The previous sentence does not make sense without a context (e.g. high, above, between, etc.).

L278: I am not sure that median latitudinal range may be considered an environmental predictor. I think that using SST here, as in the previous version, would be more appropriate.

L304: “The script to run all analyses and to create each figure is provided in the file Dataset S4”. I would change it to “is provided in the supplementary material”, since the script is not a dataset.

L309: “than in the southern Ecuador (3°S) were richness”. Please, correct typo in “where”.

L319: According to the text, and Table 2, four (not five) predictors were significant. Also, there is a comma missing after “medial latitudinal range”.

L320: Unnecessary commas after “see”.

L321: Fig. 3b-c. There is no plot 3d.

L334: Remove extra dot.

L368: “Southern” Chile?

L375: Remove extra comma after “probably”.

---

## Round 0.4 · accepted · Accept

Thank you for a very interesting paper that adds to the findings of my own research group on amphipods, polychaetes, fish and other taxa LDG globally.